# Optimal Membership Inference Bounds for Adaptive Composition of Sampled Gaussian Mechanisms

## Abstract

Given a trained model and a data sample, membership-inference (MI) attacks predict whether the sample was in the model's training set. A common counter-measure against MI attacks is to utilize differential privacy (DP) during model training to mask the presence of individual examples. While this use of DP is a principled approach to limit the efficacy of MI attacks, there is a gap between the bounds provided by DP and the empirical performance of MI attacks. In this paper, we derive bounds for the *advantage* of an adversary mounting a MI attack, and demonstrate tightness for the widely-used Gaussian mechanism. Our analysis answers an open problem in the field of differential privacy, namely the fact that membership inference is not $100\%$ successful even for relatively high budgets ($\epsilon > 10$). Finally, using our analysis, we provide MI metrics for models trained on CIFAR10 dataset. To the best of our knowledge, our analysis provides the state-of-the-art membership inference bounds.

## 1 Introduction

The recent success of machine learning models makes them the go-to approach to solve a variety of problems, ranging from computer vision (Krizhevsky et al., 2012) to NLP (Sutskever et al., 2014), including applications to sensitive data such as health records or chatbots. Access to a trained machine learning model, be it a white-box published model or through a black-box API, can leak traces of information (Dwork et al., 2015) from the training data. Researchers have tried to measure this information leakage through metrics such as membership inference (Shokri et al., 2017). Membership inference is the task of guessing, from a trained model, whether it includes a given sample or not. This task is interesting in its own right, as the participation of an individual in a data collection can be sensitive information. Furthermore, it also serves as the "most significant bit" of information: if membership inference fails, attacks revealing more information such as reconstruction attacks (Fredrikson et al., 2014; Carlini et al., 2020) will also fail. In other words, defending against membership inference attacks also defends against more advanced attacks.

The standard approach to provably defeat these membership privacy attacks is differential privacy (Dwork et al., 2006). The traditional variant of differential privacy defines a class of algorithms that apply on a database $D$ and respect a privacy budget $\epsilon$ and a probability of failure $\delta$[1]. These parameters control the divergence between the distribution of outcome of the algorithm when applied on two "neighboring" databases that only differ in one location. In particular, our mechanism of interest, the Sampled Gaussian Mechanism (SGM), makes an arbitrary function $f : X^* \to R^d$ differentially private by first sub-sampling a database of smaller size from the original database, using Poisson sampling, and then calculating $f$ on the sub-sampled database, and finally adding Gaussian noise to the outcome. Increasing the amount of injected noise provides stronger privacy guarantees, and enables to trade off between privacy and utility of the trained model. To measure the privacy of a given algorithm, researchers have developed advanced mathematical tools and notions such as Renyi differential privacy (Mironov, 2017; Abadi et al., 2016) and advanced composition theorems (Dwork et al., 2010; Kairouz et al., 2015). These tools allow us to calculate $(\epsilon, \delta)$ values for carefully designed algorithms.

---

[1]While intuitive, describing $\delta$ as the probability of failure is mathematically inaccurate .Meiser (2018)

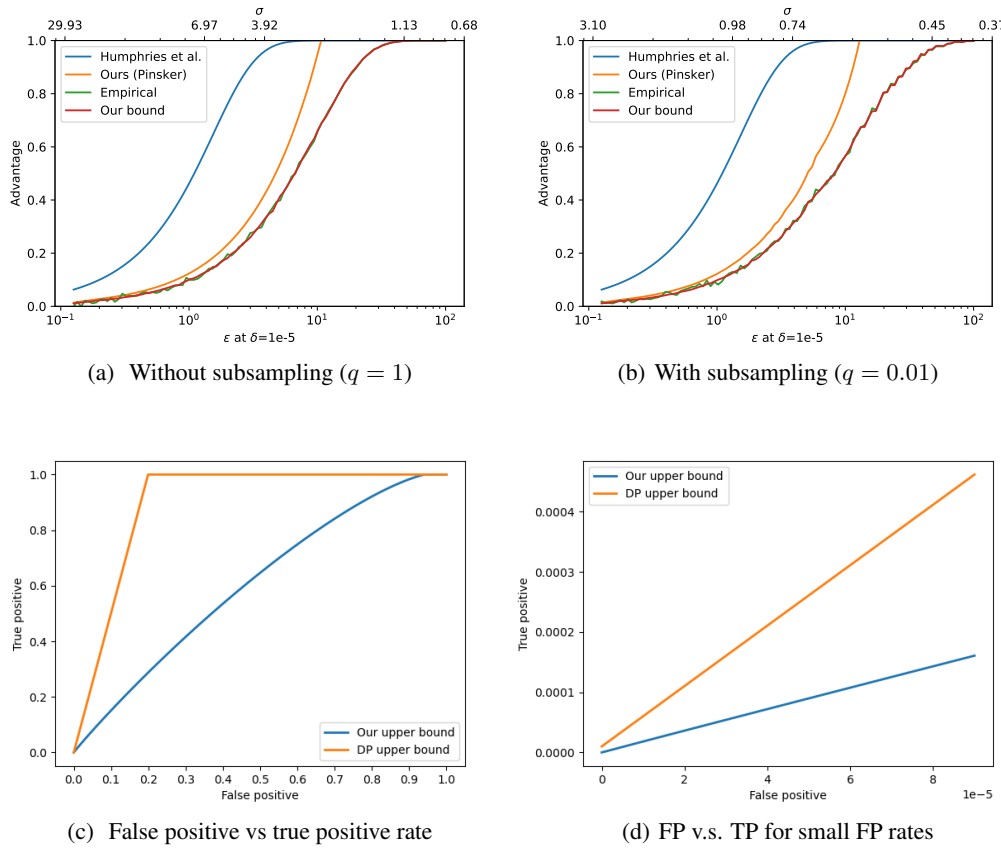

(a) Without subsampling ($q = 1$)

(b) With subsampling ($q = 0.01$)

(c) False positive vs true positive rate

(d) FP v.s. TP for small FP rates

Figure 1: Comparison of advantage bounds in a Gaussian setup. We run the sampled Gaussian Mechanism in one dimension with 50 epochs, and either no subsampling (**a**) or subsampling with $q = 0.01$ (**b**). Our attack compares likelihood of the observations under both hypothesis, and chooses the most likely (see Appendix F.1 for details). We compute the empirical advantage by measuring the percentage of time $p$ the adversary guesses membership correctly, and report $2p - 1$. In two bottom figures (c and d) we compare upper bounds for true positive rate obtained from differential privacy analysis versus our direct analysis (Details are provided in Appendix A).

Previous work has shown that any differentially private algorithm will provably bound the accuracy of any membership inference adversary. Specifically, Yeom et al. (2018), Humphries et al. (2020), and Thudi et al. (2022) prove that any model trained with $(\epsilon, \delta)$ differential privacy will induce an upper bound on the accuracy of membership inference. These upper bounds help practitioners calibrate the privacy budget $\epsilon$ to defeat the adversary. As can be seen in Figure 1, there is a large gap between the best existing upper bound and the performance of empirical attacks. In this work, we use a new proof technique to bypass DP analysis and directly bound the advantage of MI attacks, and close the gap between theoretical guarantees and empirical performance.

The most widely-used DP algorithm in machine learning applications is DP-SGD: it is a small change of the classical stochastic gradient descent algorithm that only requires to clip per-sample gradients, average them and add Gaussian noise. Each iteration of DP-SGD is an instance of the sampled Gaussian mechanism, which chooses a fraction $q$ of a dataset and outputs a noisy sum of the desired quantity. DP-SGD has been shown to be more accurate than other differentially private training algorithms in the case of linear and convex models (van der Maaten and Hannun, 2020). Our analysis mirrors the recent shift in the field of empirical membership inference, from advantage (or accuracy) metrics (Shokri et al., 2017; Yeom et al., 2018; Sablayrolles et al., 2019) to precision/recall (true positive/false positive) measures (Watson et al., 2022; Carlini et al., 2020). Our result shows that no single parameter (either $\epsilon$ or membership inference advantage) can explain the performance of adversaries in different settings. in other words, the same value for $\epsilon$ can lead to significantly different membership inference advantages and vice versa. As shown in Figure 1.c and 1.d, this phenomenon holds for the entire precision/recall curve; the precision of adversary at all given recall values cannot be explained only with a $\epsilon$ or advantage parameter.

Our contributions in this work are as follows:

- Our main theorem in Section 4 bounds the membership inference advantage of any adversary for the composition of an arbitrary set of sampled Gaussian mechanisms that could adaptively depend on each other. This bound is optimal as it reflects the membership inference advantage for a real adversary on a particular series of Gaussian mechanisms. [2] Additionally, we extend our bounds to the case where the prior knowledge of adversary is non-uniform (Appendix J). We also provide upper bounds on the true positive rate for any given false positive rate (Appendix A).

- We propose a numerical way to calculate the total variation distance (and a generalization of it) between mixtures of Gaussians (Section 4.1). Our algorithm is computationally tractable, works in linear time with respect to the number of mechanisms and is independent from the dimension. We use our numerical approach to obtain concrete bounds for Gaussian mechanisms and compare our bounds to that of Humphries et al. (2020). We also use this algorithm to calculate upper bounds on true positive rate of adversary.

- Finally, to understand the practical implication of our bound for mainstream datasets, we use DP-SGD to train models on CIFAR10 and MNIST and calculate the membership inference bounds using our techniques. Our approach allows us to achieve state-of-the-art provable membership inference privacy for any given accuracy.

## 2 BACKGROUND

In this section, we briefly recall the main properties of KL divergence and total variation (TV), as well as differential privacy and membership inference.

### 2.1 TOTAL VARIATION, KL DIVERGENCE, AND PINSKER INEQUALITY

We consider a random variable X (resp. Y) with probability distribution $\mu$ (resp. $\nu$) over a measurable space $\Omega$. With a slight abuse of notation, we consider quantities as function of either the random variables $X, Y$ or their probability distributions $\mu, \nu$. Here we recall the definition of total variation (TV) distance and Kullback-Leibler (KL) divergence:

$$\mathbf{TV}(X, Y) = \sup_{\mathcal{A}} \Pr[X \in \mathcal{A}] - \Pr[Y \in \mathcal{A}], \qquad \mathbf{KL}(X \parallel Y) = \int_{\Omega} \log\left(\frac{d\mu}{d\nu}\right) d\mu. \qquad (1)$$

**Post-processing.** Given a function $g : E \to F$, applying $g$ to $X$ and $Y$ can only decrease $\mathbf{TV}$,

$$\mathbf{TV}(g(X), g(Y)) \leq \mathbf{TV}(X, Y).$$

In particular, if $g$ is a bijective transform, we have $\mathbf{TV}(g(X), g(Y)) = \mathbf{TV}(X, Y)$.

**Pinsker's Inequality.** For any two random variables $X$ and $Y$, Pinsker (1964)'s inequality states

$$\mathbf{TV}(X, Y) \leq \sqrt{\frac{\mathbf{KL}(X \parallel Y)}{2}}. \qquad (2)$$

### 2.2 DIFFERENTIAL PRIVACY AND THE GAUSSIAN MECHANISM

**Differential privacy.** A randomized algorithm $\mathcal{M}$ satisfies $(\epsilon, \delta)$-differential privacy if, for any two datasets $D$ and $D'$ that differ in at most one sample, and any subset $R$ of the output space, we have:

$$\Pr(\mathcal{M}(D) \in R) \leq \exp(\epsilon) \Pr(\mathcal{M}(D') \in R) + \delta. \qquad (3)$$

While the probability of failure $\delta$ is often chosen to be inversely proportional to the number of samples, there is no clear consensus in the literature over desirable values of $\epsilon$.

---

[2]After submission of this work we realized (thanks to Anonymous reviewers of ICLR 3) that Theorem 4.2 in Dong et al. (2019) and its refinements in Zhu et al. (2022) implies our Theorem 6. However, since our proof technique is different, and perhaps simpler, we still keep our proof in the paper.

**DP-SGD with the subsampled Gaussian mechanism.** DP-SGD ((Song et al., 2013)) is a modification of Stochastic Gradient Descent that makes it differentially private. The core mechanism behind it is the subsampled Gaussian mechanism. Given a function $\phi$ that operates on sets $S$ with sensitivity 1 (i.e. $\|\phi(S) - \phi(S')\|_2 \leq 1$), the subsampled Gaussian mechanism selects sets $S$ at random and adds noise to the output of $\phi$. Note that the mechanism is differentially private *even* if all intermediate steps of the training process are revealed.

### 2.3 MEMBERSHIP INFERENCE ATTACKS

**Membership Inference** is the task of predicting whether a given sample was in the training set of a given model. Homer et al. (2008) showed the first proof of concept, and Shokri et al. (2017) showed that a wide variety of machine learning models are vulnerable to such attacks. Shokri et al. (2017) train neural networks to attack machine learning models, and measure the success of the attack by the percentage of correctly predicted (train/test) samples, or equivalently the advantage (Yeom et al., 2018). Later work showed that simple heuristics such as the loss (Yeom et al., 2018; Sablayrolles et al., 2019; Choquette-Choo et al., 2021) provide a more accurate and robust measure of membership inference. Membership inference attacks have also been used as an evaluation mechanism for differentially private mechanisms (Jayaraman et al., 2020; Jayaraman and Evans, 2019).

Recent works (Watson et al., 2022; Carlini et al., 2021; Rezaei and Liu, 2021) have proposed to evaluate membership inference by the precision/recall trade-off (Watson et al., 2022) or the precision at low levels of recall (Carlini et al., 2021). In particular, such works show that some setups which were thought to be private because the membership accuracy is significantly less than $100\%$ can actually reveal membership of a small group of samples with very high precision. There is also a line of work developed to design algorithms to specifically defend against membership inference attacks (Nasr et al., 2018; Jia et al., 2019; Tang et al., 2021). While these works offer no theoretical guarantees against attackers, the empirical defenses achieve better utility at the same membership advantage than their differentially private equivalents. Finally, membership inference has been studied in the context of differential identifiability (Bernau et al., 2021; Lee and Clifton, 2012) (see Appendix F.1 for Gaussian experiments that correspond to differential identifiability techniques).

## 3 MEMBERSHIP INFERENCE AND TOTAL VARIATION

In this section, we define our security game for membership inference and show its connection to the notion of total variation distance (or statistical distance) between probability distributions.

### 3.1 SECURITY GAME

We adopt the classical assumptions of membership inference (Yeom et al., 2018; Sablayrolles et al., 2019; Humphries et al., 2020). In particular, we assume that the datasets differ by adding/removing one element. We define a security game between an Adversary (who wants to guess training set membership) and a Challenger (who wants to hide training set membership).

1. Adversary picks a datasets $D = \{z_1, \ldots, z_n\}$ and a data point $z'$

2. Challenger samples a bit $b$ uniformly at random and creates

$$D' = \left\{ \begin{array}{ll} D \cup \{z'\} & \text{if } b = 1 \\ D & \text{if } b = 0 \end{array} \right.$$

3. Challenger runs the learning algorithm $L$ on $D'$ to train a model and sends a transcript of training $\theta$ to the adversary [3]

4. Adversary observes $\theta$ and guesses a bit $b'$. Adversary wins if $b' = b$.

We define the advantage of adversary $A$ on learning algorithm $L$ as $\mathbf{Adv}(L, A) = 2 \cdot \Pr[b = b'] - 1$. We also use $\mathbf{Adv}(L) = \sup_A \mathbf{Adv}(A, L)$ to denote the advantage of the worst-case adversary against $L$.

---

[3]The transcript can only include the final model or more information like the intermediate steps of training.

**Remark 1** (Comparing with security game of Yeom et al. (2018)). *The security game of Yeom et al. (2018) for membership inference attacks (See Appendix H) is slightly weaker than ours. This means that the upper bounds we prove in this work also apply to the security game of Yeom et al. (2018). A curious reader might question if our security game is too strong; the worst-case advantage of adversary might be lower in the security game of Yeom et al. (2018). In Appendix H we answer this question negatively. Through experiments, we show that we cannot prove better upper bounds in the alternative security game of Yeom et al. (2018), even if we assume the adversary does not control the neighboring datasets and only sees the final model (and not the intermediate steps). This shows that our bounds in Section 4 are tight even in the weaker security games.*

**Remark 2** (Non-uniform prior). *One can define a similar security game where the selection of the bit b is non-uniform. This security game captures settings that the adversary has some prior knowledge about the inclusion of the example in the training set. This setting was studied by Thudi et al. (2022) who provide some upper bounds for the advantage of the adversary in such a security game. In Appendix J, we show that we can extend our analysis to this setting and provide optimal bounds.*

**Comparison to existing bounds.** In the reminder of the paper we bound the membership inference advantage for DP-SGD. We first recall the best existing upper bound that is proved and improved in prior work (Jayaraman et al., 2020; Jayaraman and Evans, 2019; Humphries et al., 2020; Yeom et al., 2018; Thudi et al., 2022).

**Theorem 3** (Humphries et al. (2020)). *For any $(\epsilon, \delta)$-DP algorithm $L$ we have $\mathbf{Adv}(L) \leq \frac{e^\epsilon - 1 + 2\delta}{e^\epsilon + 1}$.*

Note that Theorem 3 applies to any DP algorithm and hence covers DP-SGD as well. Because of its generality, the bound is loose for DP-SGD (see Figure 3(a)). To improve on this bound, we perform a direct analysis of the advantage instead of using the DP tools and then converting to advantage.

**Advantage.** Let us first mathematically characterize the notion of advantage for a general learning algorithm. For a fixed learning algorithm $L$ and fixed neighboring datasets, let us define $X := \theta \mid (b = 0)$ and $Y := \theta \mid (b = 1)$ to be the distributions of the output of the learning algorithm for two cases of the security game. A deterministic adversary $A$ defines a region $\mathcal{A}$ and predicts that $\theta$ is sampled from $\mu$ if $\theta \in \mathcal{A}$ and from $\nu$ if $\theta \notin \mathcal{A}$. We use $\mu(\mathcal{A})$ and $\nu(\mathcal{A})$ to denote $\Pr[X \in \mathcal{A}]$ and $\Pr[Y \in \mathcal{A}]$. For such an adversary we have

$$\mathbf{Adv}(L, A) = \Pr[X \in \mathcal{A}] - \Pr[Y \in \mathcal{A}] = \mu(\mathcal{A}) - \nu(\mathcal{A}).$$

Note that with a simple averaging argument we can show that the best adversarial strategy in membership security game is a deterministic strategy. Therefore, the advantage for the learning algorithm $L$ is then defined as

$$\mathbf{Adv}(L) = \sup_{\mathcal{A}} \mu(\mathcal{A}) - \nu(\mathcal{A}) = \mathbf{TV}(X, Y), \tag{4}$$

where $\mathbf{TV}$ is the total variation distance. Total variation distance thus gives us an upper bound on the advantage of any adversary. Since we do not know the exact density function for $X$ and $Y$ it is not clear how to calculate or approximate the total variation distance in general. In the next subsection we discuss how Pinsker's inequality upper bounds total variation distance.

### 3.2 BOUNDING MEMBERSHIP INFERENCE USING PINSKER'S INEQUALITY

Using Equation (4) and directly applying Pinsker's inequality (2), we have:

$$\mathbf{Adv}(L) \leq \mathbf{TV}(X, Y) \leq \sqrt{\frac{\mathrm{KL}(X \parallel Y)}{2}} = \lim_{\alpha \to 1} \sqrt{\frac{D_\alpha(X \parallel Y)}{2}} \tag{5}$$

where $D_\alpha$ is the Rényi divergence at $\alpha$. This bound enables us to use Rényi divergence, which is routinely calculated in DP-SGD (Mironov, 2017) to obtain numerical upper bounds on the membership inference advantage of any adversary against adaptive composition of sampled Gaussian mechanisms.

To this end, we use RDP accounting to calculate the $D_\alpha$ for an $\alpha > 1$. We know that for any $\alpha > 1$, $D_\alpha(X, Y)$ is greater than $\mathrm{KL}(X \parallel Y)$ because $D_\alpha$ is increasing in $\alpha$. This means that for any $\alpha > 1$ $D_\alpha$ will be a valid upper bound on the membership inference advantage and the bound becomes better as we decrease $\alpha$.

In our evaluation (see Figures 3(a) and 5) we use this formulation to calculate numerical upper bounds for membership inference for DP-SGD using typical parameters. We refer to this bound as the Pinsker bound. The figure shows that it obtains better numerical values compared to the bound of Humphries et al. (2020). However, there are two main limitations with Pinsker's bound: 1) The bound is not tight. 2) It is vacuous in cases where $\mathrm{KL}(X \parallel Y) > 2$. In next section, we will optimally bound the membership inference for composition of adaptive and sampled Gaussian mechanisms.

## 4 OPTIMAL BOUNDS FOR COMPOSITION OF SAMPLED GAUSSIAN MECHANISMS

In this section, we show a tighter upper-bound for the adversary's advantage. In particular, we will upper-bound the total variation between the *transcripts* of the (subsampled) Gaussian mechanism, specifically the noisy gradients produced by the DP-SGD algorithm (Theorem 6).

As the first step of calculating our optimal bounds, we show that the adaptivity of Gaussian mechanisms does not increase the membership advantage. In other words, for clipping threshold DP-SGD with clipping threshold $r$, the entire process of DP-SGD can be replaced by a process where each step is replaced by either $\mathcal{N}(0, \sigma^2)$ or $\mathcal{N}(r, \sigma^2)$ for the basic Gaussian mechanism, and replaced by either $\mathcal{N}(0, \sigma^2)$ or $(1-q)\mathcal{N}(0, \sigma^2) + q\mathcal{N}(r, \sigma^2)$ for the sampled Gaussian mechanism, without decreasing the total variation distance. Our analysis relies on a modified version of $\mathbf{TV}$, that we denote by $\mathbf{TV}_a$.

**Definition 4.** *For $a > 0$ define $\mathbf{TV}_a(X, Y) = \frac{1}{2} \int_\Omega |\mu(x) - a \cdot \nu(x)| dx$ where $\mu$ and $\nu$ are the probability density functions for $X$ and $Y$ respectively.* [4]

While we do not know an explicit form of the $\mathbf{TV}_a$ divergence between Gaussians, we can still prove that it increases with $\|u_1 - u_2\|_2$, as formalized in the following lemma.

**Lemma 5.** *Let $X \equiv \mathcal{N}(u_1, \sigma^2 \cdot I_d)$ and $Y \equiv \mathcal{N}(u_2, \sigma^2 \cdot I_d)$. Then, for any $a \in \mathbb{R}^+$, $\mathbf{TV}_a(X, Y)$ is only a function of $\|u_1 - u_2\|_2$ and $\sigma^2$. Moreover this function is monotonically increasing with respect to $\|u_1 - u_2\|_2$. That is, for any $a \geq \|u_1 - u_2\|_2$ we have*

$$\mathbf{TV}_a(X, Y) \leq \mathbf{TV}_a(\mathcal{N}(0, \sigma^2), \mathcal{N}(a, \sigma^2)).$$

The proof involves some careful but standard gradient calculations and is deferred to Appendix B.

Now we focus on stating an upper bound on the membership inference advantage of an adversary on composition of Gaussian mechanisms. But before that, we state some useful notations. **Notation.** We use $\theta = (s_1, \ldots, s_T)$ to denote the transcript of the learning algorithm $L$ that consists of $T$ adaptive steps. We use $s_{\leq t}$ to denote the first $t$ steps of the transcript of the random process. We use $S_i$ to denote the union of the support set of the $i$th step of the mechanism $M$ on all possible datasets. That is $S_i = \{s_i; \exists D, s_i \in \mathsf{Supp}(M(D)_i)\}$. We also define $S_{\leq i} = S_1 \times \ldots, S_i$. We use $B(q)$ to denote a Bernoulli random variable that is equal to 1 with probability $q$ and 0 with probability $1 - q$. We use $0_T$ (or $1_T$) to denote the vector of dimension $T$ that is 0 (or 1) in each coordinate. We use $B(q)^n$ to denote an $n$ dimensional random variable where each coordinate is independent and distributed as $B(q)$. For $r \in \mathbb{R}$, we use $r \cdot B(q)^n$ to denote the random variable that is sampled by the following process: first sample from $B(q)^n$ and then multiply by $r$. We also use $\mathcal{N}(r \cdot B(q)^n, \sigma^2)$ to denote an $n$ dimensional random variable that is distributed according to a mixture of Gaussians which all have standard deviation $\sigma$ and centers are chosen at random from $r \cdot B(q)^n$.

**Theorem 6** (Gaussian Composition with sub-sampling). *Let $M_1, \ldots, M_T$ be a series of adaptive Gaussian Mechanisms with $L_2$ sensitivity $r$ and Gaussian noise with standard deviation $\sigma$ and sub-sampling rate $q$. The membership inference risk of the composition of $M_i$'s is at most*

$$\mathbf{TV}\big(\mathcal{N}(0^T, \sigma^2), \mathcal{N}(r \cdot B(q)^T, \sigma^2)\big)$$

*Proof sketch.* We first prove the proof for the case that there is no sub-sampling. Let $M^i(D)$ be the mechanism that works on $D$ (resp. $D'$) and consists of $i$ Gaussians $\mathcal{N}(0, \sigma^2)$ (resp. $\mathcal{N}(r, \sigma^2)$) steps followed by $T - i$ steps of SGM applied to $D$ (resp. $D'$).

Specifically, the mechanism $M^0$ corresponds to SGM, and $M^T$ to pure Gaussians. We will show in this proof that

$$\mathbf{TV}(M^i(D), M^i(D')) \leq \mathbf{TV}(M^{i+1}(D), M^{i+1}(D')) \tag{6}$$

---

[4]This notion is also used by the name of *Hockey stick divergenceBalle et al. (2018).*

and hence
$$\mathbf{TV}(X, Y) = \mathbf{TV}(M^0(D), M^0(D')) \le \mathbf{TV}(M^T(D), M^T(D')) = \mathbf{TV}(\mathcal{N}(0, \sigma^2 I_T), \mathcal{N}(r1_T, \sigma^2 I_T)).$$

To this end, we will first argue that the very final step of the mechanism $M^i$ can be replaced with a Gaussian step without increasing the total variation distance. Then we can move this noise to the start of the process without affecting the result, obtaining $M^{i+1}$.

Let us fix a step $i$ and let $s = (s_1, \ldots, s_T)$ be a transcript. Denoting $X \sim M^i(D)$ and $Y \sim M^i(D')$, we have (In the following, we use $\sum$ instead of integral for better clarity and the steps can be replace with integral.)

$$
\begin{aligned}
2\mathbf{TV}(M^i(D), M^i(D')) &= \sum_{s_{\le T} \in S_{\le T}} |\Pr[X = s_{\le T}] - \Pr[Y = s_{\le T}]| \\
&= \sum_{s_{\le T-1} \in S_{\le T-1}} \sum_{s_T \in S_T} |\Pr[X = s_{\le T}] - \Pr[Y = s_{\le T}]| \\
&= \sum_{s_{\le T-1} \in S_{\le T-1}} \sum_{s_T \in S_T} |\Pr[X_T = s_T \mid s_{\le T-1}] \cdot \Pr[X_{\le T-1} = s_{\le T-1}] \\
&\qquad\qquad - \Pr[Y_T = s_T \mid s_{\le T-1}] \Pr[Y_{\le T-1} = s_{\le T-1}]| . \\
&= \sum_{s_{\le T-1} \in S_{\le T-1}} \Pr[X_{\le T-1} = s_{\le T-1}] \sum_{s_T \in S_T} \Big| \Pr[X_T = s_T \mid s_{\le T-1}] \\
&\qquad\qquad - \Pr[Y_T = s_T \mid s_{\le T-1}] \frac{\Pr[Y_{\le T-1} = s_{\le T-1}]}{\Pr[X_{\le T-1} = s_{\le T-1}]} \Big| . \qquad (7)
\end{aligned}
$$

We know that the last ($T$'th) step of $M^i(D)$ (resp. $M^i(D')$) has two isotropic Gaussian distributions centered around two points $u_1$ and $u_2$ such that $\|u_1 - u_2\|_2 \le r$ and with standard deviation $\sigma$. These centers could be chosen adaptively according to the history of the mechanism. We use $a(s_{\le T-1})$ to denote $\frac{\Pr[Y_{\le T-1} = s_{\le T-1}]}{\Pr[X_{\le T-1} = s_{\le T-1}]}$. By Lemma 5 we have

$$
\sum_{s_T \in S_T} \Big| \Pr[X_T = s_T \mid s_{\le T-1}] - \Pr[Y_T = s_T \mid s_{\le T-1}] a(s_{\le T-1}) \Big|
$$
$$
\le 2\mathbf{TV}_{a(s_{\le T-1})}(\mathcal{N}(0, \sigma^2), \mathcal{N}(r, \sigma^2)).
$$

Denoting by $N^i$ the mechanism that coincides with $M^i$ for the first $T-1$ steps and is replaced by a Gaussian at the last ($T$'th) step, we thus have, by following Equation (7) in the reverse direction
$$2\mathbf{TV}(M^i(D), M^i(D')) \le 2\mathbf{TV}(N^i(D), N^i(D')). \qquad (8)$$
Given that the last step of $N^i$ does not depend on the first $T-1$ steps, we can permute to put it in first position.

Now, we focus on the proof of the case we have sub-sampling. We defer the full proof of Theorem 6 to the Appendix E. However, we provide a high level idea behind the proof based on the proof of the case where there is no sub-sampling. The first step is to characterize the $\mathbf{TV}_a$ for sub-sampled mechanisms and relate it to the $\mathbf{TV}_a$ for the case without sub-sampling.

**Lemma 7.** *Let $X' \equiv (1-q) \cdot Y + q \cdot X$. Then we have $\mathbf{TV}_a(X', Y) = q\mathbf{TV}_{\frac{a+q-1}{q}}(X, Y)$.*

The proof of this lemma is provided in Appendix E. Now we use Lemma 7 to convert $\mathbf{TV}_a$ of the sub-sampled Gaussians to $\mathbf{TV}_a$ of Gaussians and use Lemma 5 to upper bound them with $\mathbf{TV}_a$ of independent Gaussians. Then we use Lemma 7 again to convert the independent Gaussians to independent sub-sampled Gaussians. $\qquad\square$

**Optimality of our bound.** We note that the bound of Theorem 6 is optimal. This is because the bound is equal to the advantage of an adversary defined on sequence of independent Gaussian mechanisms. However, one might wonder if our bound is tight in the context of classification with DP-SGD. In Appendix H, we consider this question and show a learning problem for which our bound is tight. We empirically verify that advantage the adversary can get close to our upper bounds. Additionally, we show that for this learning task, an adversary who only observe the final model can also get very close to our upper bounds.

**Varying noise levels** Theorem 6 could be simply extended to composition of Gaussian mechanisms with varying noise levels, assuming the noise levels are not selected adaptively. We leave the composition of Gaussians with adaptive noise selection as an open question.

**Extending to other monotone mechanisms.** Although Theorem 6 is stated only for Gaussian mechanism, the theorem extends to any mechanism that satisfies monotonicity under $\mathbf{TV}_a$ according to some notion of sensitivity. For example, if one can show that $\mathbf{TV}_a(\mathcal{L}(0, \sigma), \mathcal{L}(u, \sigma))$ is monotonically increasing with respect to some norm of $u$, then Theorem 6 extends to composition of Laplace mechanisms with bounded sensitivity according to that norm.

### 4.1 MONTE CARLO COMPUTATION

In order to numerically approximate the upper-bound, we write the total variation as

$$\mathbf{TV}(X, Y) = \int_\Omega (d\mu(t) - d\nu(t)) \, \mathbb{1}\,(d\nu(t) \le d\mu(t)) \tag{9}$$

$$= \int_\Omega \left(1 - \frac{d\nu(t)}{d\mu(t)}\right) \mathbb{1}\,(d\nu(t) \le d\mu(t)) \, d\mu(t) \tag{10}$$

$$= \mathbb{E}_{t \sim \mu}\left(\left(1 - \frac{d\nu(t)}{d\mu(t)}\right) \mathbb{1}\,(d\nu(t) \le d\mu(t))\right). \tag{11}$$

The expectation is over the distribution $\mu$, so we can sample a dataset from $\mu$ and approximate this expectation using empirical averaging (or Monte-Carlo sampling):

$$\mathbf{TV}(X, Y) \approx \frac{1}{m} \sum_{i=1}^m \left(1 - \frac{d\nu(t_i)}{d\mu(t_i)}\right) \mathbb{1}\,(d\nu(t_i) \le d\mu(t_i)), \tag{12}$$

where $t_i \sim_{\text{i.i.d.}} \mu$. The Monte-Carlo estimation of this quantity is very accurate, as it is bounded between $0$ and $1$. Figures 3(a) and 5 show the upper bound using this Monte-Carlo simulation for typical settings in DP-SGD. We also provide a detailed algorithm on how we this in Appendix D. Following we provide a proposition on the accuracy of our Monte Carlo approximation which directly follows by applying the Chernoff-Hoefding bound.

**Proposition 8.** *Our Monte Carlo simulation (Algorithm 1 in Appendix D) returns a value $\eta$ that with high probability is close to the optimal membership inference bound $\eta^*$ of Theorem 6, namely* $\Pr[|\eta - \eta^*| \ge \delta] \le e^{-m \cdot \delta^2 / 3}$.

## 5 EVALUATION

In this section we perform experiments to calculate the concrete trade-off between accuracy and membership inference on mainstream datasets. For our experiments, we use DP-SGD (Algorithm 2) to train models and use Algorithm 1 to calculate the corresponding membership inference bounds.

**Experiments on the MNIST and CIFAR datasets.** We use the MNIST (LeCun) and CIFAR (Krizhevsky et al., 2009) datasets to perform experiments with DP-SGD. Figures 2(a) and 2(b) shows the result of our experiments. For MNIST, we used 3-layer convolutional neural networks (CNN) and train it for 10 epochs with sampling rate $q = 0.001$ and learning rate $0.1$ using the cross-entropy loss function. We also choose the gradient clipping threshold to be $1.0$ and change the noise scale between $[0.5, 1.5]$. For the CIFAR-10 experiments, we use 5-layer CNNs and trained them for 50 epochs with sampling rate $0.02$ and learning rate $0.5$ using the cross-entropy loss. We set the gradient clipping threshold to $10.0$ and the noise scale changed between $[0.5, 2]$. Our experiments show that differentially private training using DP-SGD can obtain a significantly better trade-off between accuracy and membership inference advantage than was previously thought. For instance, on the MNIST dataset we obtain membership inference bounds that are close to $0.01$ while maintaining accuracy at around $93\%$.

**Study of the role of noise scale and sampling rate.** To understand the role of sub-sampling on membership inference advantage we perform experiments with varying sub-sampling rates. In Figure 5 we show the results of our experiments where we change the sampling rate between $[0, 1]$ at three different values of noise scale. Our experiments are done for 5 epochs of training. Note that although the number of epochs are fixed, the number of iterations are different for different sampling

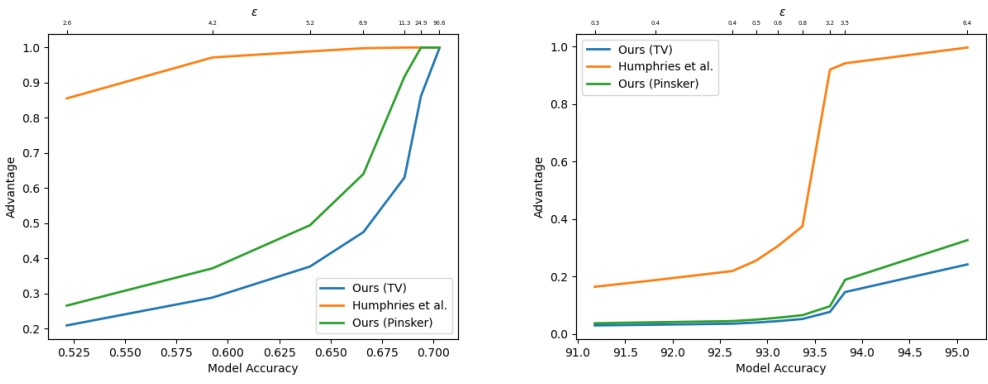

(a) Membership Inference v.s. Accuracy on CIFAR10.   (b) Membership inference v.s. Accuracy on MNIST.

Figure 2: Empirical results on membership advantage

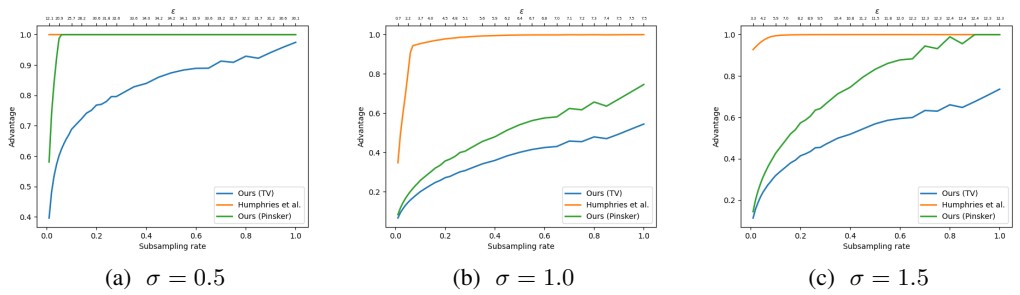

(a) $\sigma = 0.5$          (b) $\sigma = 1.0$          (c) $\sigma = 1.5$

Figure 3: Effect of sub-sampling rate at various noise scales with 5 epochs.

rates. One interesting observation is that our optimal bound is less dependent on very small sampling rate. For instance at noise scale $\sigma = 0.5$, both the Pinsker and Humphries et al. (2020) bounds become vacuous abruptly as the sampling rate grows but our optimal membership inference bound remains meaningful. Another interesting phenomenon is the oscillations that happens at certain locations for the optimal and Pinsker bounds. These oscillations happen independently for both the Pinsker and optimal bound. As these two bounds are calculated using completely different techniques, we believe this phenomenon is not due to numerical issues.

**Monte Carlo error for the experiments.** In our experiments for optimal membership bounds we use Monte Carlo simulation (see Section 4.1 and in Algorithm 1), with $m = 5e5$ samples. Via Proposition 8, we guarantee that with probability at least $99.999\%$ that the bound has error less than $0.01$.

## 6   CONCLUSION

In this paper, we directly analyzed membership inference bounds for adaptive composition of sampled Gaussian mechanisms. Our analysis enables us to obtain bounds that are better than bounds from prior analyses of differential privacy. Our analysis shows that although differential privacy guarantees might sometimes lead to large/vacuous membership inference guarantees, the mechanisms that obtain differential privacy can be in fact much more secure against membership inference attacks. Previously, this phenomenon was observed for DP-SGD and here, for the first time, we prove it.

Our analysis is the first to directly analyze membership inference bounds. We limited our study to membership inference attacks against sampled Gaussian mechanisms as DP-SGD, the most widely used differentially private learning algorithm. Direct study of membership inference could be potentially done for other mechanisms and algorithms as well. We leave this for future work.

We also note that the hyper-parameters for DP-SGD that achieve optimal membership privacy versus utility might be different than that of differential privacy. Our new analysis opens up the possibility of a systematic search for the hyper-parameters and architectures (using recent advancements in privacy-preserving hyperparameter tuning and architecture search (Papernot et al., 2020; Papernot and Steinke, 2021; Chaudhuri and Vinterbo, 2013)) that obtain optimal utility for a given upper bound on membership inference.

Last, while our paper closes the gap between theoretical and empirical measures of an adversary's advantage, the bounds apply to the subsampled Gaussian mechanism. We leave the tightness of advantage measure on other mechanisms for future work.

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

## A  BOUNDING TRUE POSITIVE RATE

As argued in introduction, membership inference alone should not be used as a single measure of privacy. Here we first show how one can use $TV_a$ to draw the false positive. Let $X, Y$ be two distributions and consider a set $S_a = \{x; \Pr[Y = x] > \frac{\Pr[X=x]}{a}\}$.

**Proposition 9.** *For all sets $S$ with $\Pr[X \in S] \le \Pr[X \in S_a]$ we have $\Pr[Y \in S] \le \Pr[Y \in S_a]$.*

*Proof.* We have

$$\Pr[Y \in S_a] - \Pr[Y \in S] = \Pr[Y \in S_a \setminus S] - \Pr[Y \in S \setminus S_a]$$

. We also have,

$$\Pr[Y \in S \setminus S_a] = \mathbb{E}_{x \leftarrow X}\Big[\frac{\Pr[Y = x]}{\Pr[X = x]} I(x \in S \setminus S_a)\Big] \le \frac{\Pr[X \in S \setminus S_a]}{a}$$

and

$$\Pr[Y \in S_a \setminus S] = \mathbb{E}_{x \leftarrow X}\Big[\frac{\Pr[Y = x]}{\Pr[X = x]} I(x \in S_a \setminus S)\Big] \ge \frac{\Pr[X \in S_a \setminus S]}{a}.$$

By the condition of the proposition we have

$$\Pr[X \in S_a] = \Pr[X \in S_a \setminus S] + \Pr[X \in S_a \cup S] > \Pr[X \in S \setminus S_a] + \Pr[X \in S_a \cup S] = \Pr[X \in S]$$

. Therefore we know that $\Pr[X \in S_a \setminus S] \ge \Pr[\mathbb{E}[X \in S \setminus S_a]$ which finishes the proof. □

**Proposition 10.** *We have*

$$\Pr[Y \in S_a] = \frac{2\mathbf{TV}_a(X, Y) + 2\Pr[X \in S_a] + a - 1}{2a}.$$

*Proof.*

$$2\mathbf{TV}_a(X, Y) = \int_\Omega |d\mu - ad\nu|$$
$$= 2\int_\Omega (ad\nu - d\mu)I(ad\nu > d\mu) + 1 - a = 2a\Pr[Y \in S_a] - 2\Pr[X \in S_a] + 1 - a.$$

□

Propositions 9 and 10 give us a way to argue about the true positive rates at any given false positive rate as follows.

**Corollary 11.** *Let $X$ and $Y$ be the distribution induced by applying composition of subsampled Gaussian mechanisms on two neighboring datasets. Also let $A$ be an adversary that given an output decides whether the output was sampled from $X$ or $Y$. Let $FP$ and $TP$ be the false positive and true positive rates of this adversary, i.e. $FP = \Pr[A(X) = 1]$ and $TP = \Pr[A(Y) = 1]$. Let $a$ be such that $\Pr_x \leftarrow X[\Pr[Y = x] > \frac{\Pr[X=x]}{a}] = FP$ Then we have $TP \le \frac{2FP + 2\mathbf{TV}_a\big(\mathcal{N}(0^T, \sigma^2), \mathcal{N}(r \cdot B(q)^T, \sigma^2)\big) + a - 1}{2a}$.*

*Proof.* This directly follows from Theorem 6 and Propositions 9 and 10. □

## B  PROOF OF LEMMA 5

*Proof.* The first part follows by the symmetry of isotropic Gaussian. For the second part (monotonicity) we use the definition of $\mathbf{TV}_a$. Without loss of generality we can assume $a \in [0, 1]$ as otherwise we can work with $\mathbf{TV}_a(P, Q)/a = \mathbf{TV}_{1/a}(Q, P)$. Let $r = \|u_1 - u_2\|_2$. We can show that the derivative of the integral is always positive. In the following calculations, we use $c_1, c_2, c_3$ and $c_4$ to denote positive constants that are independent of $r$.

First note that $x^* = \frac{r^2 - 2\sigma^2 \ln(a)}{2r}$ is a middle point where $e^{-\frac{x^2}{2\sigma^2}} - ae^{-\frac{(x-r)^2}{2\sigma^2}}$ goes from positive to negative as $x$ increases. By our assumption that $a \in [0,1]$, we have that $x^* > 0$. Recalling that $\mathrm{erf}(z) = \frac{2}{\sqrt{\pi}} \int_0^z \exp(-t^2)dt$, and that $\mathrm{erf}(\infty) = 1$ so that (by symmetry) $\frac{2}{\sqrt{\pi}} \int_{-\infty}^0 \exp(-t^2)dt = 1$, we can write

$$\mathbf{TV}_a(P,Q) = c_1 \left( \int_{-\infty}^{\infty} \left| e^{-\frac{x^2}{2\sigma^2}} - ae^{-\frac{(x-r)^2}{2\sigma^2}} \right| dx \right)$$

By breaking the integral in an intermediate point we have

$$= c_1 \left( \int_{-\infty}^{x^*} e^{-\frac{x^2}{2\sigma^2}} - ae^{-\frac{(x-r)^2}{2\sigma^2}} + \int_{x^*}^{\infty} ae^{-\frac{(x-r)^2}{2\sigma^2}} - e^{-\frac{x^2}{2\sigma^2}} \right)$$

By replacing the integrals with the CDF of Gaussian distribution we have

$$= c_1 \left( 1 + \mathrm{erf}\left( x^*/\sqrt{2}\sigma \right) - a\,\mathrm{erf}((x^* - r)/\sqrt{2}\sigma) \right)$$
$$+ \left( a(1 - \mathrm{erf}\left( (x^* - r)/\sqrt{2}\sigma \right) + (1 - \mathrm{erf}\left( x^*/\sqrt{2}\sigma \right)) \right)$$
$$= c_2 \left( \mathrm{erf}\left( \frac{r^2 - \ln(a)\sigma^2}{2\sqrt{2}\sigma r} \right) + 1 - a\,\mathrm{erf}\left( \frac{-r^2 - \ln(a)\sigma^2}{2\sqrt{2}\sigma r} \right) - a \right).$$

Now, let $f_1(r) = \mathrm{erf}\left( \frac{r^2 - \ln(a)\sigma^2}{2\sqrt{2}\sigma r} \right)$ and $f_2(r) = -a\,\mathrm{erf}\left( \frac{-r^2 - \ln(a)\sigma^2}{2\sqrt{2}\sigma r} \right)$. Taking the derivative with respect to $r$ we have

$$\frac{\partial f_1}{\partial r} = c_3 \left( \frac{1}{2\sqrt{2}\sigma} + \frac{\ln(a)\sigma}{2\sqrt{2}r^2} \right) e^{-\left( \frac{r^2 - \ln(a)\sigma^2}{2\sqrt{2}\sigma r} \right)^2}$$

$$\frac{\partial f_2}{\partial r} = c_3 a \left( \frac{1}{2\sqrt{2}\sigma} - \frac{\ln(a)\sigma}{2\sqrt{2}r^2} \right) e^{-\left( \frac{-r^2 - \ln(a)\sigma^2}{2\sqrt{2}\sigma r} \right)^2}$$

Now note that we have $e^{-\left( \frac{r^2 - \ln(a)\sigma^2}{2\sqrt{2}\sigma r} \right)^2} = a^{1/2} \cdot e^{-\left( \frac{-r^2 - \ln(a)\sigma^2}{2\sqrt{2}\sigma r} \right)^2}$. Therefore, we have

$$c_4 \frac{\partial \mathbf{TV}_a}{\partial r} = e^{-\left( \frac{-r^2 - \ln(a)\sigma^2}{2\sqrt{2}\sigma r} \right)^2} \cdot \left( \frac{1 + \sqrt{a}}{2\sqrt{2}\sigma} + \frac{\ln(a)(\sqrt{a} - 1)\sigma}{2\sqrt{2}r^2} \right).$$

Now since $a \in [0,1]$, we have $\ln(a) \le 0$ and $\sqrt{a} - 1 < 0$, which means the term $\frac{1 + \sqrt{a}}{2\sqrt{2}\sigma} + \frac{\ln(a)(\sqrt{a}-1)\sigma}{2\sqrt{2}r^2}$ is positive. This implies that the whole gradient is positive.

$\square$

## C  ESTIMATING KL WITH RÉNYI DIVERGENCE

For two probability distributions $\mu$ and $\nu$, the Rényi divergence of order $\alpha > 1$ is

$$D_\alpha(\mu \parallel \nu) \triangleq \frac{1}{\alpha - 1} \log \mathbb{E}_{t \sim \nu} \left( \frac{d\mu}{d\nu}(t) \right)^\alpha, \tag{13}$$

Rényi divergence is non-decreasing in $\alpha$, and $\lim_{\alpha \to 1} D_\alpha(P \parallel Q) = \mathrm{KL}(P \parallel Q)$.

**RDP accounting for DP-SGD.**  (Abadi et al., 2016; Mironov, 2017) propose methods to account for RDP for the Gaussian mechanism. Implementations of DP-SGD such as Opacus make use of these accounting procedures. This is important as we use these accounting methods to calculate the bound in Equation 13. Specifically, we calculate the $D_\alpha$ for $\alpha = 1 + \tau$ for a very small $\tau$, using the Opacus implementation of Rényi accounting.

## D  ALGORITHMS

---

**Algorithm 1** Optimal MI

---

**Require:** sample rate vector $q$, number of iterations $T$, learning rate vector $\eta$, noise scale vector $\sigma$, gradient norm clip vector $r$, sample size $m$

1: **for all** $j \in [m]$ **do**
2:     $\eta_j \leftarrow 0$
3: **for** $i \in [T]$ **do**
4:     **for** $j \in [m]$ **do**
5:         sample $t_{i,j} \sim \mathcal{N}(0, \sigma[i]^2 \cdot r[i]^2 \cdot \mathbb{I})$
6:         $\eta_j \leftarrow \eta_j + \ln\left(1 + q[i](e^{\frac{2t_{i,j}/r[i]-1}{2\sigma[i]^2}} - 1)\right)$
7:     $\eta \leftarrow 0$
8: **for** $j \in [m]$ **do**
9:     $\eta_j = \max(\eta_j, 0)$.
10:    $\eta \leftarrow \eta + \frac{1-e^{\eta_j}}{m}$
11: **return** $\eta$

---

**Algorithm 2** DP-SGD (Abadi et al., 2016)

---

**Require:** training dataset $D$, sample rate vector $q$, number of iterations $T$, learning rate vector $\eta$, noise scale vector $\sigma$, gradient norm clip vector $r$, loss function $L$

1: Initiate $\theta$ randomly
2: **for** $i \in \{T\}$ **do**
3:     $B_i \leftarrow$ Sample batch via Poisson sampling with rate $q[i]$
4:     $\nabla[t] \leftarrow \vec{0}$
5:     **for all** $(x,y) \in B_t$ **do**
6:         $\nabla^{(x,y)} \leftarrow$ gradient of $L(x,y)$
7:         $\overline{\nabla^{(x,y)}} \leftarrow r[t] \cdot \frac{(\nabla^{(x,y)})}{\max(r[t], \|\nabla^{(x,y)}\|_2)}$
8:         $\nabla[t] \leftarrow \nabla[t] + \overline{\nabla^{(x,y)}}$
9:     $\widetilde{\nabla[t]} \leftarrow \nabla[t] + \mathcal{N}(0, \sigma[t]^2 r[t]^2 \mathbb{I})$
10:    $\theta \leftarrow \theta - \eta\widetilde{\nabla[t]}$.
11: **return** $\theta$

---

## E    PROOF OF THEOREM 6

*Proof of Lemma 7.* We have

$$2\mathbf{TV}_a(X', Y) = \int_\Omega |d\mu' - ad\nu| = \int_\Omega |qd\mu - (q+a-1)d\nu|$$
$$= q\int_\Omega \left|d\mu - \frac{(q+a-1)}{q}d\nu\right|$$
$$= 2q\mathbf{TV}_{\frac{a+q-1}{q}}(X, Y).$$

$\square$

*Proof of Theorem 6.* The proof steps are similar to there is no sub-sampling. First, we have

$$2\mathbf{TV}(X,Y) = \sum_{s_{\leq T-1} \in S_{\leq T-1}} \Pr[X_{\leq T-1} = s_{\leq T-1}] \cdot$$
$$\left(\sum_{s_T} \left| \Pr[X_T = s_T \mid s_{\leq T-1}] - \Pr[Y_T = s_T \mid s_{\leq T-1}]\frac{\Pr[Y_{\leq T-1} = s_{\leq T-1}]}{\Pr[X_{\leq T-1} = s_{\leq T-1}]} \right| \right)$$
$$= 2\sum_{s_{\leq T-1} \in S_{\leq T-1}} \Pr[X_{\leq T-1} = s_{\leq T-1}]\mathbf{TV}_{a(s_{\leq T-1})}(X_T \mid s_{\leq T-1}, Y_T \mid s_{\leq T-1}).$$

But since $X_T$ and $Y_T$ are subsampled Gaussian mechanisms we have $X_T \equiv (1-q)Y_T + qX'_T$ where $Y$ and $X'$ are mixtures of Gaussians. Therefore, by Lemma 5 and Lemma 7 we have

$\mathbf{TV}(X, Y)$

$$
= \sum_{s_{\leq T-1} \in S_{\leq T-1}} \Pr[X_{\leq T-1} = s_{\leq T-1} \in S_{\leq T-1}] q \mathbf{TV}_{\frac{a(s_{\leq T-1})+q-1}{q}} (X'_T \mid s_{\leq T-1}, Y_T \mid s_{\leq T-1}) \text{ (By Lemma 7)}
$$

$$
\leq \sum_{s_{\leq T-1} \in S_{\leq T-1}} \Pr[X_{\leq T-1} = s_{\leq T-1} \in S_{\leq T-1}] q \mathbf{TV}_{\frac{a(s_{\leq T-1})+q-1}{q}} (\mathcal{N}(0,\sigma), \mathcal{N}(r,\sigma)) \text{ (By Lemma 5)}
$$

$$
= \sum_{s_{\leq T-1} \in S_{\leq T-1}} \Pr[X_{\leq T-1} = s_{\leq T-1}] \mathbf{TV}_{a(s_{\leq T-1})} (\mathcal{N}(0,\sigma), (1-q)\mathcal{N}(0,\sigma) + q\mathcal{N}(r,\sigma)) \text{ (By Lemma 7)}
$$

$$
= \sum_{s_{\leq T-1} \in S_{\leq T-1}} \Pr[X_{\leq T-1} = s_{\leq T-1}] \mathbf{TV}_{a(s_{\leq T-1})} (\mathcal{N}(0,\sigma), \mathcal{N}(r \cdot B(q), \sigma)).
$$

Therefore, we can replace $X_T$ with a mixture of two Gaussians centered at $0$ and $r$ and $Y_T$ with a single Gaussian centered at $0$. Now we can use the same technique used in the proof of the case without sub-sampling and move $X_T$ and $Y_T$ to the first round and repeat this process. At the end, $Y$ is replaced by a $n$-dimensional Gaussian centered at $0$ and standard deviation $\sigma$, and $X$ by a mixture of Gaussians with center randomly selected according to a $n$-dimensional Bernoulli distribution with probability $q$. That is, the advantage is bounded by

$$
\mathbf{TV}(\mathcal{N}(0^T, \sigma), \mathcal{N}(rB(q)^T, \sigma)).
$$

$\square$

# F  EXPERIMENTAL DETAILS

## F.1  GAUSSIAN EXPERIMENT

The simple Gaussian experiment is aimed at stripping away parts of the machine learning pipeline that can interfere with privacy / membership inference, such as the particularities of neural networks or optimization algorithms.

In this setup, $D = \{0, 0, \ldots, 0\}$ and $D' = D \cup \{1\}$. The (clean) summed gradient, before noise addition, is therefore either $0$ (on $D$ or on $D'$ if the batch does not contain 1) or 1 (if the batch contains 1). The adversary observes the noisy sums and infers whether they come from $D$ or $D'$. Given that the adversary knows the distribution is either $\mathcal{N}(0, \sigma^2)$ or $(1-q)\mathcal{N}(0, \sigma^2) + q\mathcal{N}(1, \sigma^2)$, they can perform a simple likelihood test to determine whether the noisy sums come from $D$ or $D'$, and predict the more likely dataset. We report the advantage of this adversary in the "empirical" curve of Figure 5.

# G  MEMBERSHIP INFERENCE PRECISION

In this section, we refine the analysis of Sablayrolles et al. (2019) for the accuracy of a membership attack.

**Upper-bound on precision.**  Let us first derive a bound on the precision of membership inference. We assume that there are two datasets $D$ and $D'$ and that a differentially-private mechanism $\mathcal{M}$ trains a model represented by $\theta$.

With probability $(1-\delta)$ over the choice of $\theta$, we have:

$$
-\epsilon \leq \log \left( \frac{\Pr(M(D) = \theta)}{\Pr(M(D') = \theta)} \right) \leq \epsilon \tag{14}
$$

Given that there is a balanced prior $\Pr(D) = \Pr(D')$, using Bayes rule, we have:

$$
\Pr(D \mid \theta) = \frac{\Pr(M(D) = \theta)\Pr(D)}{\Pr(M(D) = \theta)\Pr(D) + \Pr(M(D') = \theta)\Pr(D')} \tag{15}
$$

$$= \frac{\Pr(M(D) = \theta)}{\Pr(M(D) = \theta) + \Pr(M(D') = \theta)} \tag{16}$$

$$= \sigma \left( \log \left( \frac{\Pr(M(D) = \theta)}{\Pr(M(D') = \theta)} \right) \right), \tag{17}$$

with $\sigma(u) = 1/(1 + \exp(-u))$ the sigmoid function.

Hence the precision $\Pr(D \mid \theta)$ is bounded between $\sigma(-\epsilon)$ and $\sigma(\epsilon)$, as $\sigma(\cdot)$ is non decreasing.

**Upper-bound on attack accuracy.** The accuracy of the Bayes classifier is

$$\mathrm{Acc} = \max(\Pr(D \mid \theta), 1 - \Pr(D \mid \theta)), \tag{18}$$

and thus

$$\mathrm{Acc} \leq \max(\sigma(\epsilon), \sigma(-\epsilon)) \tag{19}$$
$$= \sigma(\epsilon) \tag{20}$$

This means that the attack accuracy is bounded by $\sigma(\epsilon)$ with probability $1 - \delta$. Empirically, we see that the sigmoid function closely matches the bound given by Humphries et al. (2020). Simply stated, this derivation shows that the bound proven by Humphries et al. (2020) actually holds with probability $1 - \delta$ instead of on average.

$$\mathrm{Acc} = \frac{1}{2} \left( \Pr(X \in \mathcal{A}) + \Pr(Y \in \mathcal{A}^c) \right)$$
$$= \frac{1}{2} \left( \Pr(X \in \mathcal{A}) + 1 - \Pr(Y \in \mathcal{A}) \right)$$
$$= \frac{1}{2} \left( 1 + \mathrm{Adv} \right)$$

## H  COMPARING SECURITY GAMES

In what follows, we write multiple variants of security games specifically defined for DP-SGD. Then we proceed to compare the security games based on an example. We perform experiments on this examples by running an attack and show that our upper bounds are tight even for the weaker security games.

### H.1  OUR SECURITY GAME WITH ALL INTERMEDIATE GRADIENTS (OIG)

1. Adversary picks a datasets $D = \{z_1, \dots, z_n\}$ and a pair of data points $z'_0, z'_1$.

2. Challenger samples a bit $b$ uniformly at random and creates

$$D' = \begin{cases} D \cup \{z'_1\} & \text{if } b = 1 \\ D \cup \{z'_0\} & \text{if } b = 0 \end{cases}$$

3. Challenger runs DP-SGD on $D'$ to train a model and sends a transcript of training, including all intermediate gradients, $\theta$ (The transcript could only include the final model or more information like the intermediate steps of training) to the adversary.

4. Adversary observes $\theta$ and and guesses a bit $b'$. Adversary wins if $b' = b$.

**Remark 12.** *In this paper we are interested in analyzing the membership inference advantage for algorithms that could be stated as adaptive composition of sampled Gaussian mechanisms. DP-SGD (Algorithm 2) is a widely used example of such algorithm. Note that we assume that the output of DP-SGD includes all the intermediate gradients that are used to train the model. In other words, the parameter $\theta$ in the security game contains all the intermediate gradients (not only the final model). However, in what follows we still define the security game for the case that the adversary only sees the final model. We define these more restricted security game so as to experimentally compare it with our security game.*

**Remark 13** (Replacing vs addition/removal). *We also note that in this section we define the notion of advantage for neighboring datasets where one dataset replaces a single example with another example (i.e. the replacement game). The reason for this choice is that the security game of Yeom et. al. is based on replacement and we want to make a fair comparison.*

*In the main body, we use the addition/removal definition of neighboring datasets because it is stronger. Namely, we can convert an upper bound on addition/removal security game to an upper bound for the replacement security game.*

**Remark 14** (Alternative notion of inference in Humphries et al. (2020)). *Humphries et al. (2020), propose a new definition for inference attacks. This definition (They call it "Experiment with Data Dependencies".) is distinct from the previously known notions of membership inference and deals with the ability of an adversary in distinguishing samples from two different distributions. As this notion is a model of distributional inference, the power of adversary in this model is not bounded by differential privacy. The only way to bound this notion of privacy with DP is to pay the cost of group privacy for groups that are almost the same size as of the entire dataset. Hence, we do not study this model in this work.*

### H.2 OUR SECURITY GAME WITH FINAL MODEL (OFM)

1. Adversary picks a datasets $D = \{z_1, \ldots, z_n\}$ and a pair of data points $z'_0, z'_1$.
2. Challenger samples a bit $b$ uniformly at random and creates

$$D' = \left\{ \begin{array}{ll} D \cup \{z'_1\} & \text{if } b = 1 \\ D \cup \{z'_0\} & \text{if } b = 0 \end{array} \right.$$

3. Challenger runs DP-SGD on $D'$ to train a model and sends the final model $\theta$ (The transcript could only include the final model or more information like the intermediate steps of training) to the adversary.
4. Adversary observes $\theta$ and and guesses a bit $b'$. Adversary wins if $b' = b$.

### H.3 YOEM ET. AL'S SECURITY GAME WITH FINAL MODEL (YFM)

1. Challenger samples a dataset $D = \{z_1, \ldots, z_{n+1}\}$ from a distribution $\mathcal{D}$.
2. Challenger runs DP-SGD on $D$ to train a model and sends a the final model $\theta$ to the adversary.
3. Challenger samples a bit $b$ uniformly at random and creates

$$z' = \left\{ \begin{array}{ll} z \leftarrow D & \text{if } b = 1 \\ z \leftarrow \mathcal{D} & \text{if } b = 0 \end{array} \right.$$

4. Adversary observes $(\theta, z')$ and and guesses a bit $b'$. Adversary wins if $b' = b$.

### H.4 YOEM ET. AL'S SECURITY GAME WITH ALL INTERMEDIATE GRADIENTS (YIG)

1. Challenger samples a dataset $D = \{z_1, \ldots, z_{n+1}\}$ from a distribution $\mathcal{D}$.
2. Challenger runs DP-SGD on $D$ to train a model and sends a transcript of training $\theta$, including all intermediate gradients, to the adversary.
3. Challenger samples a bit $b$ uniformly at random and creates

$$z' = \left\{ \begin{array}{ll} z \leftarrow D & \text{if } b = 1 \\ z \leftarrow \mathcal{D} & \text{if } b = 0 \end{array} \right.$$

4. Adversary observes $(\theta, z')$ and and guesses a bit $b'$. Adversary wins if $b' = b$.

**Notation.** For a security model $T$ we use $T(L, n, \mathcal{D})$ to denote the advantage of strongest adversary in that threat model with learning algorithm $L$, dataset size $n$ and data distribution $\mathcal{D}$ (our threat models do not use this parameter but we include it for symmetry).

**Proposition 15.** *For any learning algorithm L, any $n \in \mathbb{N}$ and data distribution $\mathcal{D}$ we have* $\mathrm{OFM}(L, n, \mathcal{D}) \leq \mathrm{OIG}(L, n, \mathcal{D})$ *and* $\mathrm{YFM}(L, n, \mathcal{D}) \leq \mathrm{YIG}(L, n, \mathcal{D})$.

**Proposition 16** (Proved in Humphries et al. (2020)). *For any learning algorithm L, any $n \in \mathbb{N}$ and data distribution $\mathcal{D}$ we have* $\mathrm{YIG}(L, n, \mathcal{D}) \leq \mathrm{OIG}(L, n, \mathcal{D})$ *and* $\mathrm{YFM}(L, n, \mathcal{D}) \leq \mathrm{YIG}(L, n, \mathcal{D})$.

**Corollary 17.** *Any upper bound on the advantage of adversaries in security game* OIG *is also an upper bound on the advantage of adversaries in the security games* YIG, YFM *and* OFM.

Our analysis above shows that upper bounds for our security model are valid upper bounds for the security game of Yoem et al and Shokri et al as well. Now, to analyze the tightness of our upper bound we perform experiments with attacks in these threat models. We argue that one cannot get a better upper bound on membership inference, unless they make extra assumptions on the data distribution. In what follows, we experimentally verify this.

## I    EXPERIMENTS ON THE OPTIMALITY OF OUR BOUND

In this section, we construct a data distribution and study logistic regression on this distribution.

**Definition 18.** *We define $\mathcal{H}_d$ be the uniform distribution over the hamming ball of radius 1 and centered at zero. Thus, the samples from $\mathcal{H}_d$ have the form, $(0, \ldots, 0, 1, 0, \ldots, 0)$ with only one of the coordinates being 1 and other being 0. For an arbitrary Boolean function $f \colon \{-1, 1\}^d \to \{0, 1\}$ we also define $\mathcal{H}_d^f$ to be the distribution of samples from $\mathcal{H}$ that are labled w.r.t. $f$, namely, $\mathcal{H}_d^f = (\mathcal{H}_d, f(\mathcal{H}_d))$.*

**Experiment setup:**    We run experiments for membership inference on DP-SGD when trying to learn $\mathcal{H}_d^f$, using logistic regression, and for an arbitrary function $f$. We set the learning rate to 0.001, the clipping threshold to .1 and vary the sub-sampling rate and noise multiplier. We run the models for either 5 or 50 epochs.

**Attacks.**    We implement a simple attack that only looks at the final model. This adversary looks at the final model $\theta$ and the target instance $x$ in hand which is equal to 1 in coordinate $i$ and zero everywhere else. If the $i$th coordinate of $\theta$ is larger than $Tqc/2 + mTqc/d$ ($T$ is the number of iterations, $q$ is sub-sampling rate, $n$ is the number of examples in the training set, and $c$ is the clipping threshold) then the attack predicts $b' = 1$ otherwise it predicts $b' = 0$. We call this attack the "final model attack" (FMA). We also implement another attack that looks at all the intermediate models. This attack basically performs the FMA attack at each iteration and takes majority vote at the end. We call this attack the "Intermediate models attack" (IMA).

**Evaluation.**    We evaluate our FMA attack in the threat models of Yoem et al. and Shokri et al. in the setting where the adversary only sees the final model. We also evaluate this attack in our threat model, in the setting that the adversary only sees the final model. We also report the accuracy of the stronger IMA attack.

### I.1    RESULTS

We now summarize our findings in our experiments

**Increasing $d$ reduces the gap between threat models.**    We instantiate our data distribution $\mathcal{H}_d^f$ with a random function $f$ and select the dimension from $\{2000, 10000, 10000\}$. We set the sample size to 1000, set the sub-sampling rate to .1 and vary the noise multiplier to obtain the attack curve. We run the models for 5 epochs and report the attack in various threat models. Our results in Figure I.1 show that by increasing the dimension, the gap between the performance of all attacks and our upper bound shrinks to almost 0. This verifies the optimality of our bound. It also shows that there is no fundamental gap between the threat models and we cannot hope to achieve stronger upper bounds in the weaker threat models, unless we make further assumptions.

**Small sub-sampling rate results in a small gap between the performance of FMA and IMA.** We instantiate our data distribution $\mathcal{H}_d^f$ with a random function $f$ and set dimension to 2000 while keeping the number of instances at 1000. We set the sub-sampling rate to .01 and the number of epochs to 50. We vary the noise multiplier to obtain the attack curve. Our results in Figure I.1 show

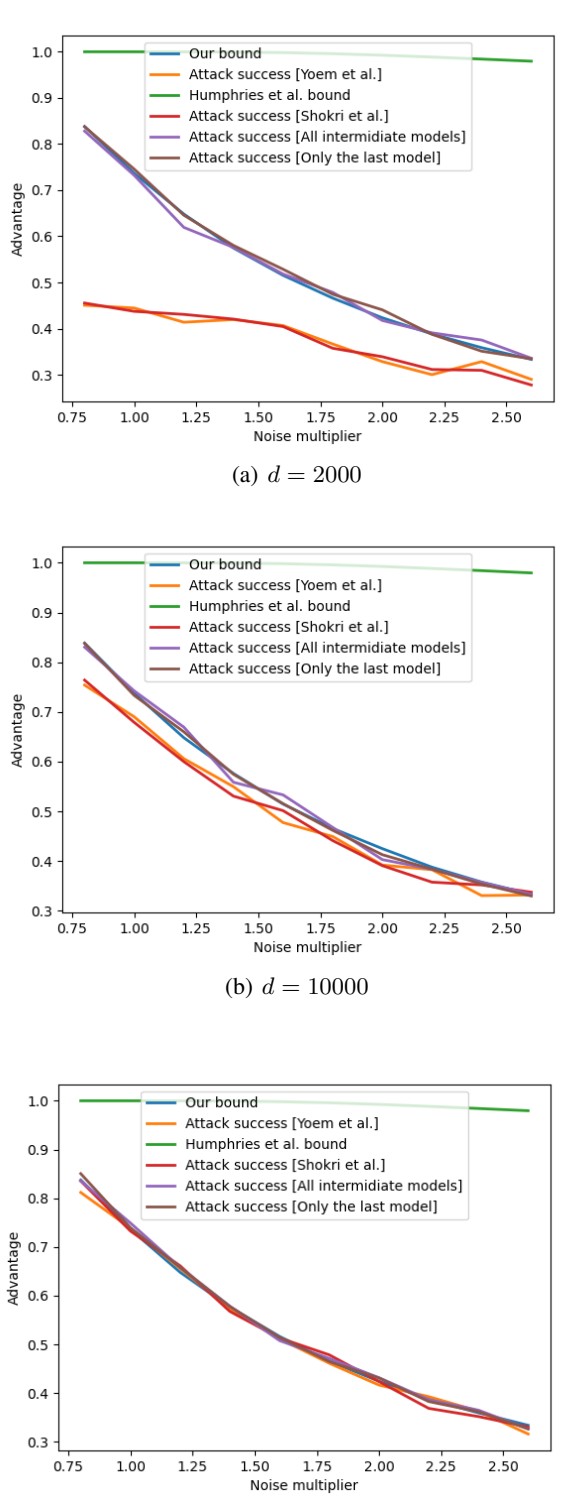

(a) $d = 2000$

(b) $d = 10000$

(c) $d = 100000$

Figure 4: Decreasing $d$ results in smaller gap between all threat models and the upper bound. This shows the optimality of our bound in all threat models.

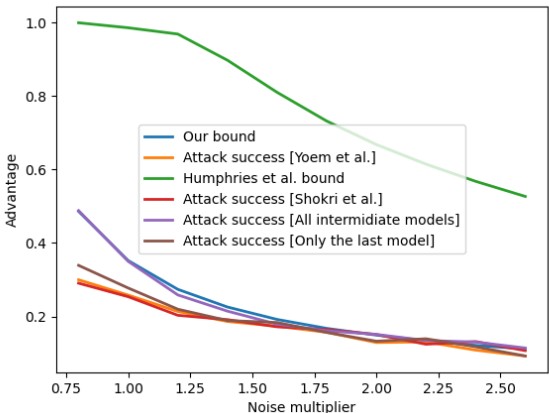

Figure 5: Small sampling rate can produce a gap between FMA and IMA, but this gap tightens as we increase the noise.

that small sub-sampling rate can create a gap between the performance of two attacks. This gap shrinks as we increase the noise multiplier.

**Remark 19.** *Our experiments in this section are verification of the observation made in Nasr et al. (2021) that an adversary with enough capabilities (e.g. poisoning the dataset and observing the models in different iterations) can achieve high membership inference advantage. Our result in this section show that if we allow the adversary to pick the classification problem (the data distribution and the architecture), it can achieve high advantages that are close to theoretical upper bounds, even without seeing the intermediate models or poisoning the data.*

## J  SECURITY GAME WITH NON-UNIFORM PRIOR

Here, we extend our result to the setting where the prior distribution for the bit $b$ in the security game for membership inference is non-uniform. This setting is recently studied in the work of .

**Definition 20** (Non-uniform membership inference.). *We define a security game between an Adversary (who wants to guess training set membership) and a Challenger (who wants to hide training set membership).*

1. *Adversary picks a datasets $D = \{z_1, \ldots, z_n\}$ and a data point $z'$*

2. *Challenger samples a bit $b$ from a bernouli distribution with probability $p$ and creates*

$$D' = \left\{ \begin{array}{ll} D \cup \{z'\} & \text{if } b = 1 \\ D & \text{if } b = 0 \end{array} \right.$$

3. *Challenger runs the a learning algorithm $L$ on $D'$ to train a model and sends a transcript of training $\theta$ (The transcript could only include the final model or more information like the intermediate steps of training) to the adversary.*

4. *Adversary observes $\theta$ and guesses a bit $b'$. Adversary wins if $b' = b$.*

*We define the advantage of adversary $A$ on learning algorithm $L$ as $\mathbf{Adv}(L, A, p) = 2 \cdot \Pr[b = b'] - 2 \max(p, 1 - p)$. We also use $\mathbf{Adv}(L, p) = \sup_A \mathbf{Adv}(A, L, p)$ to denote the advantage of any adversary against $L$.*

Note that similar to the uniform setting, with a simple averaging argument we can show that the best adversarial strategy in the non-uniform membership security game is a deterministic strategy. Therefore, assuming $p < 0.5$, the advantage for the learning algorithm $L$ is then defined as

$$\frac{\mathbf{Adv}(L, p)}{2} = \sup_{\mathcal{A}} \mu(\mathcal{A}) \cdot p + (1 - \nu(\mathcal{A})) \cdot (1 - p) - (1 - p) \tag{21}$$

$$= \sup_{\mathcal{A}} (1-p) \left( \nu(A) - \frac{p}{1-p} \mu(A) \right) \tag{22}$$

$$= \sup_{\mathcal{A}} (1-p) \left( \nu(A) - \frac{p}{1-p} \mu(A) \right) - (1-p) \left( \nu(\bar{A}) - \frac{p}{1-p} \mu(\bar{A}) \right) \tag{23}$$

$$+ (1-p) \left( \nu(\bar{A}) - \frac{p}{1-p} \mu(\bar{A}) \right) \tag{24}$$

$$= 2(1-p) \mathbf{TV}_{\frac{p}{1-p}}(X, Y) + (1 - 2p - \frac{\mathbf{Adv}(L, p)}{2}). \tag{25}$$

Therefore we have

$$\mathbf{Adv}(L, p) = 2(1-p) \mathbf{TV}_{\frac{p}{1-p}}(X, Y) + 1 - 2p \tag{26}$$

Now using this, we can prove the following Theorem.

**Theorem 21** (Non-uniform gaussian Composition with sub-sampling). *Let $M_1, \ldots, M_T$ be a series of adaptive Gaussian Mechanisms with $L_2$ sensitivity $r$ and Gaussian noise with standard deviation $\sigma$ and sub-sampling rate $q$. The non-uniform membership inference risk of the composition of $M_i$'s is at most*

$$2(1-p) \mathbf{TV}_{\frac{p}{1-p}} \left( \mathcal{N}(0^T, \sigma), \mathcal{N}(r \cdot B(q)^T, \sigma) \right) + 1 - 2p$$

*where $p < 0.5$ is the probability of sampling of the additional example in the non-uniform security game.*

*Proof.* The proof is similar to the proof of Theorem 6 except that we use Equation 26 instead of Equation 4. $\square$

