# OpenReview forum: "Optimal Membership Inference Bounds for Adaptive Composition of Sampled Gaussian Mechanisms"
_ICLR.cc/2023/Conference — Submitted to ICLR 2023_

### Official Review · Reviewer_919G · 2022-10-25

**Confidence:** 4
**Correctness:** 3
**Technical Novelty And Significance:** 2
**Empirical Novelty And Significance:** 2
**Recommendation:** 5

**Clarity, Quality, Novelty And Reproducibility:**

The paper is mostly clear and very nice to read, although some points woiuld benefit from clarifications.
As far as I know, the results are novel and seem very interesting. It should be possible to replicate the main findings of the paper given the information provided.


**Strength And Weaknesses:**

### Update after the discussions:

As pointed out by the other reviews, many of the main results in the paper can be derived by existing methods. I am therefore lowering my score. I still think that this can make a nice paper once the authors have more time to rewrite it and change the focus to emphasize the more novel aspects. Hopefully, the more critical reviewers can also give good feedback on what they think would be the possible novelty.

### Strengths:

1) The results presented in the paper seem exiting, and solve an important open problem in privacy-preserving machine learning.

2) The paper is mostly enjoyable to read.

### Weaknesses:

With superficial checking of the proofs (due to tight reviewing schedule), I have no major problems with this paper. The following are minor comments and some questions for the authors:

1) Considering Remark 1 about the weaker security game: how does this relate to Nasr et al. 2021 results with DP, which shows that to achieve the theoretical DP bounds the adversary needs the full power allowed by the DP definitions, in particular the power to poison the data on each iteration and to see the output from all iterations, not only the final model?

2) Add some more steps to proof of Lemma 5 to make it easier to follow; by quick checking there also might be some typos in the proof, please check this again.

3) Thm.6: Please clarify the proof: does the transcript of the learning algorithm need to be discrete (e.g. Eq.8)? What do you mean by saying that M^i(D) "works on D and consists of i Gaussians N(0,sigma^2) steps.."?

4) Fig.4: synced oscillations look really weird, please check again that this is not an artifact of the experiment.


#### Even smaller comments:

i) Thm.7: Thm statement is overlapping with the proof.

ii) p.4: denote by Adv(L) "advantage of any adversary". Should this be advantage of worst-case adversary? It looks like there might be adversaries which have worse advantage, since this is sup over A.

iii) What is Thm.18 in Appendix D? The notation is a bit confusing: in Lemma 8 X' is a mixture of X and Y, but in the proof of Thm18 X is mixture of X' and Y.

iv) Note that delta in approximate DP is not a probability (see e.g. Meiser 2018).

v) p.3: DP-SGD was proposed by Song et al. 2013, while the main contribution by Abadi et al. 2016 was the new accounting technique.


#### References:

Meiser 2018: Approximate and probabilistic DP definitions.

Nasr et al. 2021: Adversary instantiation.

Song et al. 2013: SGD with DP updates.


**Summary Of The Paper:**

The paper focuses on finding bounds on the success of membership inference (MI) attacks, i.e., attacks that predict if a given sample has been used in training a model or not, when the privacy is protected by (subsampled) Gaussian mechanism from the differential privacy (DP) literature. MI attacks have an intimate connection with DP, since DP definitions direcly give worst-case bounds for MI. The current paper establishes optimal bounds for the adversary advantage for the subsampled Gaussian mechanism using the add/remove neighbourhood granularity. The authors experimentally show that their bounds are tighter than the existing bounds in the literature, and that the optimal bound matches the experimental results.


**Summary Of The Review:**

A very nice paper (partly) solving an open problem in privacy-preserving machine learning.

---

> ### Author Response · Authors · 2022-11-19
> **Rebuttal**
>
> We thank the reviewer for their valuable and positive comments.
>
> > Considering Remark 1 about the weaker security game: how does this relate to Nasr et al. 2021 results with DP, which shows that to achieve the theoretical DP bounds the adversary needs the full power allowed by the DP definitions, in particular the power to poison the data on each iteration and to see the output from all iterations, not only the final model?
>
> Our experiments in Appendix I are indeed complementary to Nasr et al. 2021’s analysis. We show that if we allow the adversary to pick the classification problem (the data distribution and the architecture), it can achieve high advantages that are close to theoretical upper bounds, even without seeing the intermediate models or poisoning the data. We added this remark in Appendix I.
>
> > Add some more steps to proof of Lemma 5 to make it easier to follow; by quick checking there also might be some typos in the proof, please check this again.
>
> We added two additional steps in the middle of calculation to clarify.
>
> > Thm.6: Please clarify the proof: does the transcript of the learning algorithm need to be discrete (e.g. Eq.8)?
>
>
> Indeed, this is an important point. We do not want the transcript to be discrete. We use sigma instead of integral for more clarity. We clarify this in the beginning of the proof.
>
> > What do you mean by saying that M^i(D) "works on D and consists of i Gaussians N(0,sigma^2) steps.."?
>
> We define some artificial mechanism M^i(D) that starts with i steps of pure gaussian noise  and then continues with the T-i steps of the actual mechanisms. We similarly define the same for M^i(D’) but instead of gaussian noise centered at 0 we use gaussian noise centered at 1.
>
> > Fig.4: synced oscillations look really weird, please check again that this is not an artifact of the experiment.
>
> Indeed, this was surprising to us too. Upon further investigation, we found the source for this oscillation. Note that we fix the number of epochs to 5, and then set the number of iterations to int(5/q) where q is the sampling rate. For this reason, when q goes from 0.8 to 0.81, the number of iterations reduces from 6 to 5. This is the main reason we observe these weird oscillations. Thank you for pointing this out.
>
> Thank you for all the other comments. We addressed them all. We also want to note that we have now included additional experiments for calculating the optimal precision recall curve in Figure 1 ( and how to calculate them in Appendix A). We would really appreciate any further comment on these new figures.

---

### Official Review · Reviewer_8xNP · 2022-10-25

**Confidence:** 4
**Correctness:** 4
**Technical Novelty And Significance:** 2
**Empirical Novelty And Significance:** 2
**Recommendation:** 5

**Clarity, Quality, Novelty And Reproducibility:**

My main criticism is regarding the novelty: I would claim that using existing tools, it is already possible to bound the TV distance tightly for the subsampled Gaussian mechanism (and many more), using the results of [3] and also using the numerical method of [2]. Namely, the way you define TV, i.e. $\mathrm{TV}(X,Y) = \sup_A \mathbb{P}(X \in A) - \mathbb{P}(Y \in A)$, we see that this is just the hockey-stick divergence for $\varepsilon=0$ i.e. the $\alpha$-divergence for $\alpha=1$ (see e.g. [1]). The methods of [2] and [3] can be used to compute this for the subsampled Gaussian mechanism with arbitrary precision. When writing the hockey-stick divergence using the privacy loss distribution, we just need to compute an integral from $\varepsilon$ to $\infty$ and having $\varepsilon=0$ does not affect the method at all. It is of course nice if there are analytical formulas for doing it, however these computational methods are very efficient and thus I think this reduces the novelty of this paper a lot.

You seem to define $\mathrm{TV}(X,Y)$ without the absolute value, however it would not change anything as for $(\varepsilon,\delta)$ we take maximum of $H_\alpha(X,Y)$ and $H_\alpha(Y,X)$, where $H_\alpha$ denotes the $\alpha$-divergence.


I think that the TV-results for the Gaussian mechanism and the simple formula for the subsampling amplification for TV are nice contributions, but I don't think they are sufficient for a publication.

[1] Balle, B., Barthe, G., & Gaboardi, M. (2018). Privacy amplification by subsampling: Tight analyses via couplings and divergences. Advances in Neural Information Processing Systems, 31.

[2] Gopi, S., Lee, Y. T., & Wutschitz, L. (2021). Numerical composition of differential privacy. Advances in Neural Information Processing Systems, 34, 11631-11642.

[3] Zhu, Y., Dong, J., & Wang, Y. X. (2022, May). Optimal accounting of differential privacy via characteristic function. In International Conference on Artificial Intelligence and Statistics (pp. 4782-4817). PMLR.


**Strength And Weaknesses:**


Pros:

- the paper improves the TV bounds by Humphries et al. (2020) for the subsampled Gaussian mechanism considerably.

- The paper is very well written, easy to follow.

Cons:

- There is not much novelty, the contribution remains small (see comments below).

**Summary Of The Paper:**

The paper gives formulas for evaluating the total variation distance for the outputs of the Gaussian mechanism and its subsampled variant (i.e. for outputs originating from neighbouring datasets). The need for doing this is motivated by membership inference security game where an adversary tries to guess whether a given sample is in the data or not. The 'advantage' of the adverary can be bounded using the total variation distance. This has been considered previously for general $(\varepsilon,\delta)$-DP mechanisms by Humphries et al. (2020).

**Summary Of The Review:**

Doe to these aforementioned reasons (lack of novelty, lack of content), I am leaning towards rejecting the paper.

---

> ### Author Response · Authors · 2022-11-19
> **Rebuttal**
>
> Thank you for your valuable comments.
> We admit that we were not aware of the notion of trade-off (characteristic) function at the time of our submission.  After carefully reviewing the Roth e al and Zhu et al, we agree with the reviewer that our theorem 6 is covered by these works. We have now added a footnote explaining this. We also shortened our proof and only state the proof for a sub-sampled case. We note that we still keep the proof in the paper because 1) our proof technique is different and 2) our non-uniform bounds in Appendix J  use this proof. We want to clarify a few other points about our result:
>
>
>
> - We want to emphasize that our proof technique is different and could be useful in analyzing other mechanisms. For instance, as we clarify in a paragraph titled “Extending to other monotonous mechanisms”, we can extend this proof to any mechanism that incurs a “dominating” pair of distributions.
> - We emphasize that one of the main novelties of our work is to directly analyze membership inference. Although our analysis is not as novel as we previously thought it was, our goal is still novel and has not been done before.
> - We also want to emphasize that we are the first to show the large gap between analyzing membership inference directly vs analyzing it using epsilon delta values. As pointed out by Reviewer 919G, this solves a big open question in the context of membership inference.
> - Our analysis also can obtain the optimal false positive versus true positive curve. Note that our theorem 6  is true for any value of $a$ in $TV_a$ (hockey stick divergence). This means we can use it to obtain the optimal precision recall curve. We now have made this connection clear in Appendix A. We also added plots for this curve in Figure 1. The curve shows that the upper bound on true positive rate
>
> -We also clarify that our bounds in the non-uniform setting are new and are not covered by the RDS paper.
>
>
> Regarding numerical calculation using the result of [2], We believe they need epsilon to be non-negligible. Their running time depends on 1/eps and is not efficient for small epsilon. Nonetheless, our numerical analysis takes a completely different approach of using monte-carlo approximation which we think is an interesting contribution on its own. Moreover, our monte carlo approach allows us to calculate the entire precision/recall curve in a very efficient manner.

---

> > ### Comment · Reviewer_8xNP · 2022-11-22
> > **Thank you for the rebuttal**
> >
> > Thanks for acknowledging my comments. I agree with some of them, but I would also argue that the trade-off function for any pair of random variables (and alsothe optimal false positive versus true positive curve) can be obtained via the hockey-stick divergence (Zhu et al. results, it is given there). Also, I think that, despite the theoretical running time guarantees of [2], numerically the hockey-stick divergence can be evaluated very fastly for $\epsilon=0$. I think the paper has its merits, but is not sufficient for publication and thus I keep my score.

---

### Official Review · Reviewer_eNLu · 2022-10-28

**Confidence:** 4
**Correctness:** 3
**Technical Novelty And Significance:** 2
**Empirical Novelty And Significance:** Not applicable
**Recommendation:** 3

**Clarity, Quality, Novelty And Reproducibility:**

The paper is quite poorly and hastily written.
The idea of directly reasoning about the Total Variation distance when reasoning about the advantage is novel.

**Strength And Weaknesses:**

Strengths:
1. This paper studies a relevant problem.
2. The paper makes headway in the right direction by trying to bound the advantage directly using the divergences which are intimately tied to DP guarantees rather than incurring the additional loss incurred by DP guarantees by converting them to divergence based guarantees.

Weaknesses:
1. The paper is poorly and hastily written; many statements are imprecise and its unclear what they mean. I will list down a few below, and would be interested to know what the authors mean by them:
 *  On page 2, "Our analysis explains, from a theoretical point of view, why the precision and recall of membership inference attacks is stronger than the accuracy." --- What is accuracy here? How does your analysis explain this? There is no mention of precision/recall after the related work, its unclear what the paper is claiming.
* Notation of r is used before defining in section 4, page 5 when giving an overview of the section. There are also other such instances.  While this by itself doesn't make a paper rejection worthy, it creates quite an unpleasant experience.
* Throughout the paper, there has been reference to Theorem 18, when the real reference is to something else. This again, makes reading and trying to understand the contributions really hard.
2. The equation following Lemma 5 doesn't show a monotonically increasing function, while maybe not wrong, the equation is quite vacuous. Also, the proof is quite standard.
3. For Theorem 6, while I agree with the statement, reading the proof is more confusing than just trying to reason about the statement.
4. The optimality claim is very vague, there is no rigorous sense in which optimality is defined and shown. If it is provably optimal, one doesn't need to empirically show tightness; if it is not, it is not optimal.

**Summary Of The Paper:**

The paper studies upper bounds on the performance of membership inference (MI) attacks on models trained with the subsampled Gaussian mechanism to ensure differential privacy. In this paper, the authors derive bounds on the advantage of an adversary mounting the MI attack.They also perform some experiments to illustrate the values of the calculated upper bounds on CIFAR and MNIST datasets.

**Summary Of The Review:**

All in all, the paper is not very well written and makes it hard to understand. In light of the weaknesses mentioned above, I recommend this paper for rejection.

---

> ### Author Response · Authors · 2022-11-19
> **Rebuttal**
>
> Thank you for your valuable comments. Below, we try to answer your questions.
>
>
> > What is accuracy here? How does your analysis explain this? There is no mention of precision/recall after the related work, it’s unclear what the paper is claiming.
>
> We agree that this text was rather confusing. We have now revised this text and highlighted it in red. We have also added new results about precision/recall (true positive/ false positive) rate for membership inference adversaries (See Figure 1 for these curves and also explanations on how we calculate this in appendix A).
>
> >Notation of r is used before defining in section 4, page 5 when giving an overview of the section. There are also other such instances. While this by itself doesn't make a paper rejection worthy, it creates quite an unpleasant experience.
>
> We updated the draft and added the meaning of $r$ before using it.
>
> > Throughout the paper, there has been reference to Theorem 18, when the real reference is to something else. This again, makes reading and trying to understand the contributions really hard.
>
> We apologize for this reference issue. It is now fixed. Also note that we have merged the theorem for sub-sampling and without sub-sampling.
>
> >The equation following Lemma 5 doesn't show a monotonically increasing function, while maybe not wrong, the equation is quite vacuous. Also, the proof is quite standard.
>
> We have now clarified the statement of the theorem. We also mentioned that the proof follows from standard gradient calculations.
>
> > The optimality claim is very vague, there is no rigorous sense in which optimality is defined and shown.
>
> We refined the optimality paragraph. Note that our bounds are optimal because the worst-case bound is “realizable” through composition of gaussian mechanisms that are independent. Our experiments are only performed to show that the worst case bounds could be realized by applying DP-SGD on a classification task. We also have experiments for more restricted adversarial settings.

---

> > ### Comment · Reviewer_eNLu · 2022-12-06
> > **Response**
> >
> > Thank you for your response to my comments.
> > The changes do make the exposition clearer. But I still don't think this paper has enough novel content to warrant a full paper submission.
> > I would like to keep my score.

---

### Official Review · Reviewer_H7BJ · 2022-10-31

**Confidence:** 4
**Correctness:** 3
**Technical Novelty And Significance:** 3
**Empirical Novelty And Significance:** Not applicable
**Recommendation:** 3

**Clarity, Quality, Novelty And Reproducibility:**

A membership inference (MI) attack is a hypothesis test between the null hypothesis H_0 (the person is not in the dataset) and the alternative hypothesis H_1 (the person is in the dataset. A complete understanding of what an MI attack can do is given by the tradeoff between type I error (or false positive rate) and type II error (false negative rate). Let $T(\alpha)$ be the minimum type II error of any attack that has type I error at most $\alpha$. The TV distance between the two hypotheses, which is the advantage measure studied in the paper under review, is then simply $\max_{\alpha \in [0,1]} 1 - \alpha - T(\alpha)$. Thus, the tradeoff curve $T$ allows computing the TV distance, but also gives more fine grained information about false positive vs false negative tradeoffs.

The paper of Dong, Roth, and Su mentioned above has a detailed study of this tradeoff curve. They show how to bound a tradeoff curve of any adaptive composition of mechanisms by the tradeoff curve of a non-adaptive composition: see Theorem 4 in the DRS paper. Applying Theorem 4 from DRS to DP-SGD immediately yields Theorem 7 from the paper under review, and, more strongly, it yields a complete tradeoff curve between type I and type II error. Moreover, Theorem 10 from the DRS paper already gives this analysis of DP-SGD. So, as far as I understand, the theoretical results in this paper are subsumed by the DRS results.

It may be also worth noting that the TV_a quantity is commonly known as the hockey-stick divergence, and Lemma 5 is also known: see the results in the ICML 2018 paper by Balle and Wang.

Finally, the notation in the paper is unclear in some places. What is, for example, $1_T$ in (6)? What is $0^T$ in Theorem 7?

**Strength And Weaknesses:**

A detailed understanding of membership inference attacks is an important research agenda. This paper studies a widely used differentially private algorithm, and gives tight bounds on the advantage over random guessing allowed by this algorithm. There are, however, some shortcomings to this work:

* The theoretical results are special cases of the much more general results of Dong, Roth, and Su ("Gaussian Differential Privacy", Journal of Royal Statistical Society, Series B). In particular, the DRS results hold for a wider range of algorithms, and allow a more fine grained understanding of the tradeoff between true and false positives in an inference membership attack. I elaborate on this below.

* The advantage over random measure is a crude way to measure the power of an attack. The authors themselves acknowledge that precision and recall are better measures in evaluating attacks. But then what is the value of bounding advantage for a particular algorithm?

* The model where the whole history of noisy gradient updates is available to to an attacker is not justified, and not always realistic. It may make sense in some federated learning settings, but the authors need to justify their model.

* The analysis works only for a single, very specific algorithm.

**Summary Of The Paper:**

Motivated by membership inference attacks against differentially private stochastic gradient descent (DP-SGD), the paper bounds the total variation (TV) distance between two runs of DP-SGD on two neighboring data sets. DP-SGD amounts to subsampling and adding Gaussian noise to adaptively chosen vectors. The main result of the paper is that the TV distance between these two sequences of random vectors can be bounded by a sequence where the vectors are fixed rather than adaptively chosen, or, equivalently, by the TV distance between two Gaussian distributions with equal variance and random means chosen from a binomial distribution. The result then implies bounds on the advantage over random guessing of a membership inference attacks against DP-SGD, assuming that the attacker sees the full sequence of noisy gradient steps.

**Summary Of The Review:**

The paper studies an important topic but the results follow from prior work, and have some considerable limitations.

---

> ### Author Response · Authors · 2022-11-19
> **Rebuttal**
>
> We thank the reviewer for their comments.
>
>
> We admit that we were not aware of the notion of trade-off function at the time of our submission.  After carefully reviewing the RDS, we agree with the reviewer that our theorem 6 is covered by theorem 4.2 in RDS. We have now added a footnote explaining this. We also shortened our proof and only state the proof for a sub-sampled case. We note that we still keep the proof in the paper because 1) our proof technique is different and 2) our non-uniform bounds in Appendix J  use this proof. We want to clarify a few other points about our result:
>
>
>
> - We want to emphasize that our proof technique is different and could be useful in analyzing other mechanisms. For instance, as we clarify in a paragraph titled “Extending to other monotonous mechanisms”, we can extend this proof to any mechanism that incurs a “dominating” pair of distributions.
> - We emphasize that one of the main novelties of our work is to directly analyze membership inference. Although our analysis is not as novel as we previously thought it was, our goal is still novel and has not been done before.
> We also want to emphasize that we are the first to show the large gap between analyzing membership inference directly vs analyzing it using epsilon delta values. As pointed out by Reviewer 919G, this solves a big open question in the context of membership inference.
> - Our analysis also can obtain the optimal false positive versu true positivel curve. Note that our theorem 6  is true for any value of $a$ in $TV_a$ (hockey stick divergence). This means we can use it to obtain the optimal precision recall curve. We now have made this connection clear in Appendix A. We also added plots for this curve in Figure 1. The curve shows that the upper bound on true positive rate is much better than the one obtained by differential privacy.
> - We also want to note that the RDS paper does not provide a way to calculate the exact value of the trade-off function. They have a theorem showing that one can *approximate” the curve with provable convergence when the number of mechanisms go to infinity. However, the convergence rate is unknown and there is not guarantee on the quality of approximation. Our monte carlo approach in approximating $TV_a$ is novel in its own right.
> - We also clarify that our bounds in the non-uniform setting (Appendix J) are new and are not covered by the RDS paper.
>
>
>
> All in all, thank you very much for bringing this work to our attention.
>
> >The advantage over random measure is a crude way to measure the power of an attack.
>
> Thank you for your comment. We have now added false_positive/ true_positive curves based on our analysis. Appendix A shows how we calculate these bounds based on the notion of tv_a and we plot the results in figure 1.
>
> > The model where the whole history of noisy gradient updates is available to to an attacker is not justified, and not always realistic. It may make sense in some federated learning settings, but the authors need to justify their model.
>
> The reason we choose this model is that it is the strongest adversarial setting and proving upper bounds for this setting would be stronger. If reviewer's concern is about the optimality claims, in Appendix I (and also Remark 1), we argue that using other possible threat models where the adversary only sees the final model (and also does not see other training data) would not improve the upper bounds. We show that although this setting might seem to be much weaker, the adversary can still obtain membership inference advantage close to our upper bound if we allow the adversary to choose the data distribution. This shows that we cannot obtain stronger bounds for the membership inference advantage in these weaker threat models unless we make assumptions on the data distribution.
>
> > The analysis works only for a single, very specific algorithm.
>
> DP-SGD is the mainstream algorithm used for differentially private machine learning and it is one of the only known algorithms that can obtain provable membership inference privacy. We hope that our approach of directly analyzing membership inference can motivate this kind of analysis for other membership inference defenses.
>
> >  What is 0_T
>
> By $0_T$ we mean a vector of size T with 0 in all coordinates.

---

### Decision · Program_Chairs · 2023-01-20

**Decision:**

Reject

**Justification For Why Not Higher Score:**

The work is of little novelty. The main results mostly follow as special cases of a previous work.

**Justification For Why Not Lower Score:**

N/A

**Metareview: Summary, Strengths And Weaknesses:**

The paper studies membership inference attacks against DP-SGD algorithm. In particular, the authors analyze the total variation (TV) distance between two executions of this algorithm on two neighboring datasets.

Understanding membership inference attacks is an important goal, however, the paper has several shortcomings. The general consensus among the reviewers is that there is little novelty in this work. In particular, several results in the paper follows from a prior work ([Dong et al. 2019]) as pointed out by one of the reviewers. Moreover, the results are based on analyzing a very specific instantiation of one algorithm and under specific assumptions on the adversary's knowledge, which limit their applicability.